# E/e’ Ratio Predicts the Atrial Pacing-Induced Left Atrial Pressure Response in Patients with Preserved Ejection Fraction

**DOI:** 10.3390/medicina59020210

**Published:** 2023-01-21

**Authors:** Seung-Young Roh, Dae-In Lee, Kwang-No Lee, Jinhee Ahn, Yong-Soo Baek, Dong-Hyeok Kim, Jaemin Shim, Jong-Il Choi, Young-Hoon Kim

**Affiliations:** 1Division of Cardiology, Department of Internal Medicine, Korea University College of Medicine and Korea University Medical Center, Seoul 02841, Republic of Korea; 2Division of Cardiology, Department of Internal Medicine, Chungbuk National University Hospital, Chungju-si 28644, Republic of Korea; 3Department of Cardiology, Ajou University School of Medicine, Suwon 16499, Republic of Korea; 4Division of Cardiology, Department of Internal Medicine, Pusan National University Hsopital, Busan 49241, Republic of Korea; 5Division of Cardiology, Department of Internal Medicine, Inha University Hospital, Incheon 22332, Republic of Korea; 6Division of Cardiology, Department of Internal Medicine, Ewha University Hospital, Seoul 07804, Republic of Korea

**Keywords:** diastolic dysfunction, heart failure, atrial fibrillation, atrial hypertension, left atrial pressure

## Abstract

*Background and Objectives:* Left atrial hypertension is one of the pathophysiologies of heart failure with preserved ejection fraction. We hypothesized that left atrial pressure response (LAPR) to incremental pacing is higher in patients with atrial fibrillation (AF) and can predict left ventricular diastolic dysfunction. *Materials and Methods:* Patients requiring left atrial access as a part of a therapeutic procedure for AF (*n* = 204, AF group) or supraventricular tachycardia (*n* = 34, control group) were analyzed (male *n* = 183, 54 ± 12 years old). LAPR was measured during incremental pacing. *Results:* Baseline left atrial pressure and LAPR at all pacing rates were not different between the AF and control groups. They were higher in patients with a high E/e’ (≥ 8) than in those with a low E/e’ (<8). LAPR at a pacing interval of 400 ms and E/e’ were positively correlated (r = 0.373, *p* < 0.001). Body mass index and a high E/e’ were independent predictors of pacing-induced left atrial hypertension. *Conclusions:* LAPR to incremental pacing was constant regardless of AF. The non-invasive echocardiographic marker E/e’ reflected pacing-induced left atrial hypertension.

## 1. Introduction

Heart failure (HF) is one of the leading causes of hospitalization and cardiovascular mortality regardless of the left ventricular (LV) ejection fraction (EF). Several studies have demonstrated similar mortality rates between individuals having HF with preserved EF (HFpEF) and those having HF with reduced EF [1,2]. The main hemodynamic pathophysiology of HFpEF is the elevation of LV filling pressure (LVFP). In patients with advanced HFpEF, LVFP is elevated at rest. However, in the early stage, increased LVFP is observed only during strenuous physical activity [3]. High LVFP during exercise in HFpEF is associated with symptoms such as dyspnea and aerobic capacity reduction. If HFpEF progresses over time, left atrial (LA) remodeling and dysfunction develop. 

Thus, LA remodeling reflects the cumulative effects of elevated LVFP. Elevated LA pressure (LAP) is related to LA remodeling in the general population regardless of atrial fibrillation (AF). It provides diagnostic and prognostic information about LV diastolic dysfunction and the chronicity of the disease. A recent study showed that elevated LAP is associated with extended electro-anatomical remodeling of the LA and poor clinical outcomes after AF ablation [4]. In addition, it is known to trigger AF by causing ectopic beats emanating from the pulmonary veins (PVs) [5]. However, invasively measured LAP is insufficient to identify the stage of HFpEF because it is usually not increased in the early stage because of LA adaptation. In addition, it is not a fixed parameter because it is sensitive to body volume and heart rate [6].

Therefore, we hypothesized that the LAP response (LAPR) to incremental pacing reflects LV diastolic dysfunction. The objective of this study was to reveal the clinical implications and non-invasive predictors of LAPR.

## 2. Methods

### 2.1. Study Population

We screened a population who needed LA access as a part of a therapeutic procedure for AF or supraventricular tachycardia (Figure 1). From July 2015 to November 2016, 264 patients with AF were enrolled. The control group consisted of 35 patients with re-entry tachycardia via a left-side accessory pathway or left-origin atrial tachycardia. Patients with (1) previous cardiac surgery or procedure history (*n* = 0), (2) LV systolic dysfunction (LVEF < 50%) or structural heart disease including ischemic lesion (*n* = 15), (3) moderate to severe mitral and aortic valve disease (*n* = 0), (4) recurrent triggers, that induced sustained arrhythmias interrupting the maintain sinus rhythm (SR) (*n* = 1), and (5) AF induction during right atrial pacing (*n* = 14 in AF group and *n* = 1 in control group) were excluded via a screening test. A total of 204 patients in the AF group and 34 patients in the control group were finally analyzed (male 77.1%, 54.0 ± 12.4 years old). In addition, the cohort was divided into two groups based on the criteria of E/e’ = 8 (median value), which is an echocardiographic LV diastolic dysfunction marker. We compared 144 patients with low E/e’ and 124 patients with high E/e’. All patients provided written informed consent for inclusion in the cohort. The research protocol complied with the principles of the Declaration of Helsinki and was approved by the Institutional Review Board of the Korea University Anam Hospital.

### 2.2. Echocardiography

All candidates underwent transthoracic echocardiography within a month prior to the procedure. Cardiac chamber size, LV wall thickness, blood flow velocity, and tissue Doppler images of the mitral annular septal region were assessed. The measurement of E and e’ criteria followed the current guidelines [7]. The E wave represents the ratio of peak velocity of blood flow from left ventricular relaxation in early diastole. The E wave was measured with optimal alignment of the PW Doppler sample between mitral leaflet tips. e’ is a measure of peak mitral annular PW Doppler velocity at lateral and septal basal regions during early filling, so average e’ velocity is recorded. Of the 268 enrolled patients, 164 had SR and 104 had AF at the time of transthoracic echocardiography.

### 2.3. Measurement of LAP and Incremental Pacing

All patients fasted eight hours prior to invasive LAP measurement and catheter ablation. Saline fluid (0.9%) was injected during that time at a rate of 40 mL/hour without taking medication. Sedation was used during the procedure and pressure measurements. We did not use general anesthesia. Intracardiac echocardiography and measurement of hemodynamics were performed using a Prucka CardioLab electrophysiology recording system (General Electric Medical System Inc., Milwaukee, WI, USA). A septal puncture was performed to assess the LA. Systemic anticoagulation was initiated with intravenous heparin, maintaining an active coagulation time of 300 to 350 s immediately before septal puncture. A Swartz left 1 long sheath (St. Jude Medical, Inc., Minnetonka, MN, USA) was used for septal puncture. To measure LAP, a 6-F pigtail catheter (A & A Medical Devices Inc., Ansan-si, Republic of Korea) was inserted into the LA through the long sheath. Baseline LAP was measured during SR at the height of the v wave. If AF was sustained at the initial time of the procedure, SR was restored with internal cardioversion (Physio-Control Lifepack 12, Physio-Control Corp., Redmond, WA, USA) with 5–20 J of energy and LAP was measured 5 min after restoring SR. To increase the heart rate, incremental right atrial pacing was performed. The LAPRs at heart rates of 60, 75, 100, 120, and 150 beats per minute (bpm) were observed. If the patient’s breathing was unstable, LAP was measured during inspiration.

### 2.4. Statistical Analyses

Statistical analyses were performed using the SPSS 20.0 software package (SPSS Inc., Chicago, IL, USA). Continuous variables are expressed as mean ± standard deviation. They were compared by Student’s t-test, Mann–Whitney U-test, and ANOVAs, followed by post hoc analyses using Bonferroni’s method. Categorical variables are reported as counts with percentages and were compared using a Chi-square test or Fisher’s exact test. The difference in pacing-dependent LAP changes was determined by ANOVA. Multivariate analysis was conducted with a logistic regression model reporting odds ratios (ORs) to predict high LAPR (LAP ≥ 26 mmHg). Predictor variables included age, female sex, hypertension, diabetes mellitus, atrial fibrillation, body mass index, LA volume index, LV mass index, LVEF, and high E/e’. Multiple regression analysis was performed using the criterion of *p* < 0.10 in the univariate analyses for a variable to enter the model. AF-free survival was measured by the Kaplan–Meier survival curve analysis, and the difference between both groups was assessed by a log-rank test. A *p*-value < 0.05 was considered statistically significant.

## 3. Results

### 3.1. Baseline Characteristics

The baseline characteristics of the AF and control groups are presented in Appendix A. The AF group was older (56 ± 11 vs. 39 ± 14 years, *p* < 0.001) and included more men (81 vs. 59%, *p* = 0.007). This group had a higher BMI (25 ± 3 vs. 23 ± 3 kg/m^2^, *p* = 0.003) and more hypertension (38 vs. 6%, *p* < 0.001). In echocardiographic data, the AF group had a larger LV diastolic diameter (47 ± 4 vs. 45 ± 4 mm, *p* = 0.026), higher LV mass (161 ± 30 vs. 129 ± 29 g, *p* < 0.001), larger LA volume index (30 ± 6 vs. 22 ± 4 mL/m^2^, *p* < 0.001), and higher E/e’ (8 ± 3 vs. 7 ± 2, *p* = 0.003). The LVEF was preserved, and there were no significant differences between the two groups. Additionally, we analyzed the difference between patients with high E/e’ and low E/e’ (Appendix B). The high E/e’ group was older (58 ± 11 vs. 50 ± 12 years, *p* < 0.001), included fewer men (67 vs. 88%, *p* < 0.001), and had a greater number of hypertensive (45% vs. 24%, *p* < 0.001) and diabetic (13% vs. 4%, *p* = 0.005) patients than the low E/e’ group. The high E/e’ group had a more common history of myocardial infarction (3% vs. 0%, *p* = 0.045). In echocardiographic data, the high E/e’ group had a higher LV mass index (91 ± 16 vs. 85 ± 16 g/m^2^, *p* = 0.001), a larger LA volume index (31 ± 7 vs. 28 ± 6 mL/m^2^, *p* < 0.001), and higher pulmonary artery systolic pressure (30 ± 6 vs. 28 ± 6 mmHg, *p* = 0.002). Baseline LAP was not significantly different between the high E/e’ group and the low E/e’ group.

### 3.2. Left Atrial Pressure Response 

The heart rate changed according to the right atrial pacing (RAP) interval (Figure 2). LAP did not increase at 75 bpm, 100 bpm, and 120 bpm but rose at 150 bpm (27 ± 5 mmHg in the AF group; 25 ± 4 mmHg in the control group). A similar pattern was observed in both the AF and control groups. LAP was not different at any of the heart rates in both groups. In the high E/e’ group and the low E/e’ group, the LAP differed according to RAP (29 ± 6 mmHg vs. 25 ± 4 mmHg) at 150 bpm. There was a significant difference at all pacing rates. In addition, the difference was more pronounced at 400 ms than at baseline (11% vs. 16%). Both LAPs at baseline and 400 ms were related to E/e’ and positively correlated (r = 0.230 and r = 0.373) (Figure 3). Linear regression was performed to find the predictor of high LAPR (LAP ≥27 mmHg) (Table 1). BMI, LV mass index, and high E/e’ (*p* < 0.10 in the univariate analysis) were included in the multivariate analysis. BMI (OR = 1.098 [1.006–1.197], *p* = 0.035) and high E/e’ (OR = 2.054 [1.235–3.416], *p* = 0.006) were independent predictors of high LAPR.

### 3.3. Predictor of Atrial Fibrillation Recurrence

The presence of RAP-induced high LAPR could not predict AF recurrence after catheter ablation (log-rank *p* = 0.299) (Figure 4A). AF freedom outcome was not different between the high and low LAPR groups (log-rank *p* = 0.299) (Figure 4B). It also did not differ between the high and low E/e’ groups (log-rank *p* = 0.541)

## 4. Discussion

LAP increased with incremental atrial pacing. The LAPR was not correlated with the presence of AF but was closely related to E/e’, an echocardiographic marker of LV diastolic dysfunction. 

### 4.1. The Mechanism of LA Hypertension 

LAP is influenced by several factors such as LV systolic and diastolic function, LA chamber stiffness, and intravascular volume status. If LV diastolic function worsens, LV end-diastolic pressure (LVEDP) increases to maintain adequate LV stroke volume [8]. The LA is directly exposed to LV pressure during its diastolic phase. The filling of the LA during the LV systolic phase produces high LAP, which leads to increased LA wall tension and remodeling. LAP reflects both LA remodeling in chronically increased LVEDP exposure and pressure loading through the mitral valve. It can be used to investigate the prognosis of HF. Increased LAP causes electrical heterogenicity of the atrial myocardium, which causes AF [9,10]. AF is a result as well as an aggravating factor of HF. We hypothesized that LAP and LAPR were increased in patients with AF, but there was no significant difference in the results. First, the degree of diastolic dysfunction was not significantly different between the AF and control groups. E/e’ was statistically different, but the absolute values were not clearly different. Many subjects with relatively mild HF were included because patients with AF are candidates for ablation. Second, the dilated and remodeled atria compensate for the pressure change.

### 4.2. The Clinical Implication of LAPR

LAP does not increase at the normal range of heart rate but increases rapidly beyond its threshold. This means that insufficient time to fill the appropriate volume leads to an increase in LVEDP. In our study, LAPR was closely related to E/e’ regardless of the rhythm status. These results imply that the main factor of LAPR is LV diastolic dysfunction rather than atrial remodeling. LV diastolic function determines the boundary value of the LAPR. Particularly, it is helpful in identifying the cause in patients who complain of non-ischemic exercise-related dyspnea. Increased LAP during exercise or tachycardia causes dyspnea, and E/e’ can be a marker of exertional dyspnea of cardiac origin. At rest, patients with diastolic dysfunction may have a cardiac output or filling pressure similar to that of healthy individuals who have a normal diastolic function. Exercise echocardiography [11] is usually performed to detect reduced LV systolic and/or diastolic reserve capacity in the setting of coronary disease or diastolic dysfunction. The result of exercise echocardiography can be predicted using E/e’, which closely reflects LAPR.

### 4.3. The Clinical Implication of E/e’ as a Marker of Early LV Diastolic Dysfunction

E/e’ measured by echocardiography is a non-invasive method that reflects LV diastolic dysfunction. LAPR is a marker of LV diastolic reservoir, but it has to be obtained using an invasive method. E/e’ measured by echocardiography is a non-invasive method and was found to be closely correlated with the LAPR [12]. Several other studies have shown a good correlation between E/e and pulmonary capillary pressure or LV mean diastolic pressure during variable levels of exercise [13,14]. E/e’ is clinically useful regardless of rhythm status. 

Baseline LAP, LAPR, and E/e’ were not able to predict the outcome after catheter ablation. The most relevant predictor of prognosis after catheter ablation is LA remodeling, including enlargement and fibrosis. The main determinant of both LAP and LAPR is LV diastolic dysfunction rather than LA remodeling. This study included only people with relatively mild HF. In other studies, increased E/e’ was a predictor of poor outcomes after ablation, such as low LA voltage [15]. As a result, it is impossible to determine a candidate for ablation considering E/e’ in the early stages of HF. However, it can be helpful in deciding pre- and post-procedural medication, and what causes mainly provoke dyspnea.

### 4.4. Study Limitations

The following limitations should be considered when interpreting this result. This was a single-center observational study that included only patients selected for catheter ablation of AF or supraventricular tachycardia. Therefore, it is difficult to generalize this finding to the entire population. Since most of the patients in the AF group had compensated for HF and proper general condition, there may not be any difference from the control group. This tends to ignore the effect of atrial remodeling on LAP. Next, the difference in rhythm status should be considered when LAP was measured. In patients with persistent AF, LAP was measured after returning to SR after cardioversion; however, it may not have recovered from stunning even after 5 min. However, this result was consistently observed in other patients who were measured without cardioversion. Lastly, LAP measured during tachycardia induced by pacing may differ from that during increased heart rate due to exercise and emotional changes in daily life. This is because the increase in heart rate by activity is accompanied by an increase in LV contractility, aortic stiffness, and preload in response to an increase in sympathetic tone. For this reason, it is difficult to mention that the results of this study perfectly reflect the heart response during ordinary exercise.

## 5. Conclusions

In conclusion, LAP showed a constant increase with the heart rate change according to pacing. The echocardiographic non-invasive marker, E/e’, reflected the LAPR measured during incremental pacing. It can be an indication to evaluate the cause of exertional dyspnea regardless of AF.

## Figures and Tables

**Figure 1 medicina-59-00210-f001:**
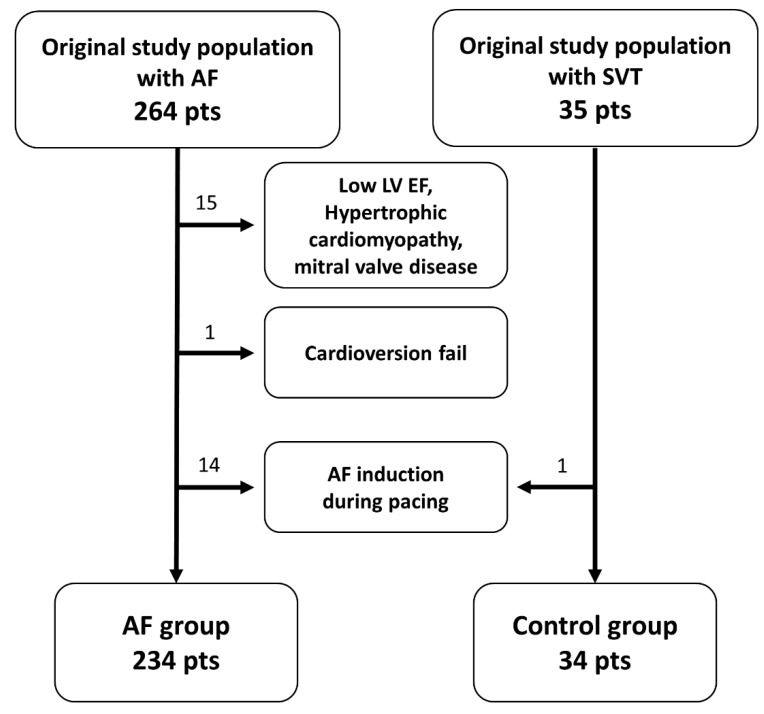
The diagram demonstrates patient enrollment and disposition during the study; AF, atrial fibrillation; SVT, supraventricular tachycardia; LV, left ventricular; EF, ejection fraction.

**Figure 2 medicina-59-00210-f002:**
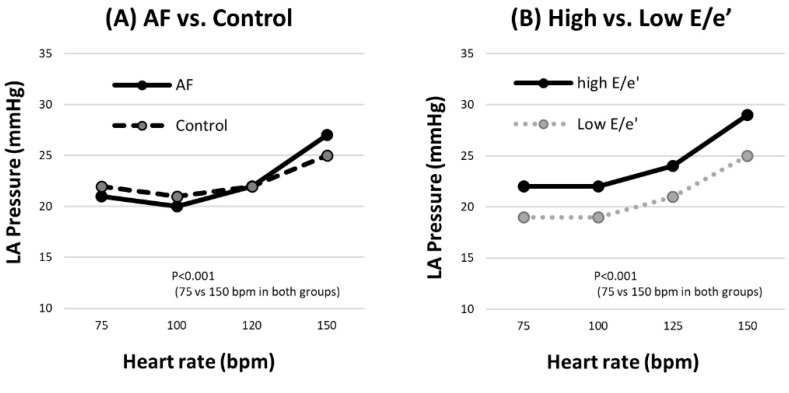
The pattern of heart rate dependent LA pressure response. (**A**) In AF and control groups, LA pressure response as heart rate change showed a similar pattern. (**B**) The different LA pressure response between both low and high E/e’ group. AF, atrial fibrillation; LA, left atrial.

**Figure 3 medicina-59-00210-f003:**
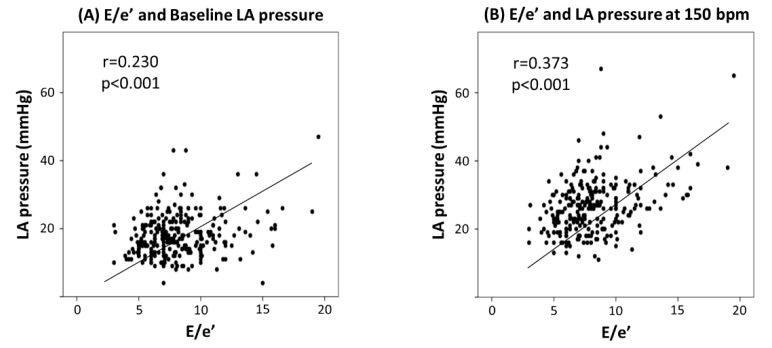
Correlation between left atrial pressure and E/e’ (**A**) at baseline and (**B**) at a heart rate of 150 bpm. LA, left atrial.

**Figure 4 medicina-59-00210-f004:**
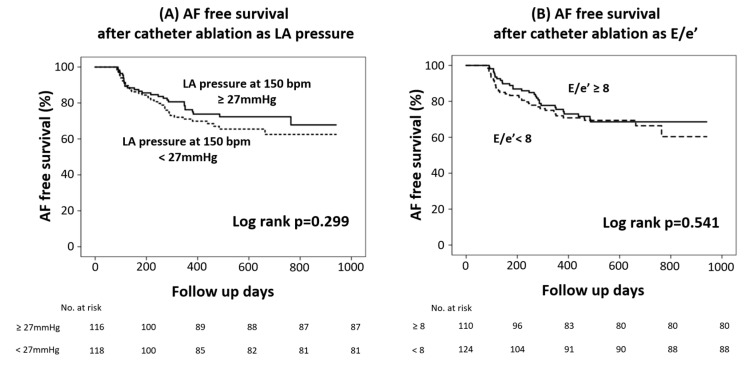
Atrial fibrillation/atrial tachycardia free survival as a clinical outcome after catheter ablation. (**A**) High (≥27 mmHg at 150 bpm) versus low (<27 mmHg at 150 bpm) left atrial pressure response group. (**B**) High (E/e’ ≥ 8) versus low (E/e’ < 8) E/e’ group. AF, atrial fibrillation; AT, atrial tachycardia; LA, left atrial.

**Table 1 medicina-59-00210-t001:** Predictors of high LA pressure response (LA pressure >26 mmHg) on univariate and multivariate analysis.

Variables	Univariate OR (95% CI)	*p*-Value	Multivariate OR (95% CI)	*p*-Value
Age, 1 year	1.013 (0.993–1.033)	0.198		
Female sex	1.085 (0.609–1.934)	0.782		
HTN	1.279 (0.769–2.126)	0.343		
Diabetes mellitus	0.610 (0.244–1.525)	0.291		
AF	0.904 (0.440–1.859)	0.784		
Body mass index, kg/m^2^	1.111 (1.021–1.208)	0.014	1.098 (1.006–1.197)	0.035
LA volume index, mL/m^2^	1.026 (0.981–1.073)	0.213		
LV mass index	1.018 (1.003–1.034)	0.017	1.014 (0.998–1.197)	0.081
LV ejection fraction, %	1.069 (0.975–1.173)	0.153		
High E/e’ (≥8)	2.336 (1.430–3.818)	0.001	2.054 (1.235–3.416)	0.006

AF, Atrial fibrillation; LV, Left ventricular; LA, Left atrial; NT Pro BNP, N terminal brain natriuretic peptides.

## Data Availability

The data presented in this study are available on request from the corresponding author. The data are not publicly available due to privacy restrictions.

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
