# Peer review of "E/e’ Ratio Predicts the Atrial Pacing-Induced Left Atrial Pressure Response in Patients with Preserved Ejection Fraction"

_medicina, 2023, doi:10.3390/medicina59020210_

Round 1

Reviewer 1 Report

I have read the paper: “E/e’ ratio predicts the atrial pacing-induced left atrial pressure 2 response in patients with preserved ejection fraction”. It is a retrospective analysis on predictors of left atrial hypertension induced by pacing. The manuscript is well written.

Some comments:

     1)     “Methods”: exclusion criteria. Could please the authors clarify if aortic valve disease was an exclusion criterion? As also disease of aortic valve like for the mitral valve may increase left atrial pressure.  

      2)     “Methods”: echocardiography. Please clarify how both E and e’ were calculated. Indeed, e’ changes from lateral to medial mitral annulus and guidelines recommend considering the mean. Please cite the echocardiography guidelines that were used as appropriate.

      3)     LAP also depends on the volume (intravascular volume or preload) of the patient. How the authors accounted for variations in preload and how they standardized for changes in intravascular volume among patients?

      4)     Kaplan Meier curves (Figure 4). Number at Risk Tables should be provided. In panel A patients with low LA pressure at 150 bpm seems to have less AF recurrence. The not significant p-value might be secondary to low power (small cohort of patients). However in absence of Risk Tables it is difficult to draw any conclusion.

      5)     Kaplan Meier curves (Figure 4). Why did the author stratify only for LA pressure at 150 bpm. What are the results for LA pressure dichotomized at other pacing cycle length? Any significant difference in AF free survival?

Author Response

I have read the paper: “E/e’ ratio predicts the atrial pacing-induced left atrial pressure 2 response in patients with preserved ejection fraction”. It is a retrospective analysis on predictors of left atrial hypertension induced by pacing. The manuscript is well written.

We appreciate the many comments and suggestions provided by reviewer #1. In addressing them, the manuscript has been significantly improved.

Some comments:

“Methods”: exclusion criteria. Could please the authors clarify if aortic valve disease was an exclusion criterion? As also disease of aortic valve like for the mitral valve may increase left atrial pressure.  

Thank you for this valuable comment. We wanted to measure the relationship between left ventricular diastolic function and left atrial pressure change in people with ordinary ventricular systolic function. That's why people with structural heart disease were excluded. Moderate to severe aortic valvular disease is also classified as structural heart disease and included in the exclusion criteria. However, no patients with moderate or higher aortic valve disease have been screened for study in practice. We have changed statement in manuscript to clarify.

Page 2 line 77

(3) moderate to severe mitral valve disease

à moderate to severe mitral and aortic valve disease

2) “Methods”: echocardiography. Please clarify how both E and e’ were calculated. Indeed, e’ changes from lateral to medial mitral annulus and guidelines recommend considering the mean. Please cite the echocardiography guidelines that were used as appropriate.

Thank you for this comment. We added the measurement of E and e' to the echocardiography section of the method part. Guidelines for measurement methods have been added as references.

Page 3 line 98

The E wave represents the ratio of peak velocity of blood flow from left ventricular relaxation in early diastole. e’ is a measure of peak mitral annular velocity during early filling.

à The measurement of E and e’ criteria followed the current guidelines. The E wave represents the ratio of peak velocity of blood flow from left ventricular relaxation in early diastole. E wave was measured with optimal alignment of PW Doppler sample between mitral leaflet tips. e’ is a measure of peak mitral annular PW Doppler velocity at lateral and septal basal regions during early filling. So average e’ velocity is recorded.

  1. Nagueh, S.F.; Smiseth, O.A.; Appleton, C.P.; Byrd, B.F., 3rd; Dokainish, H.; Edvardsen, T.; Flachskampf, F.A.; Gillebert, T.C.; Klein, A.L.; Lancellotti, P.; et al. Recommendations for the Evaluation of Left Ventricular Diastolic Function by Echocardiography: An Update from the American Society of Echocardiography and the European Association of Cardiovascular Imaging. Eur Heart J Cardiovasc Imaging 2016, 17, 1321-1360, doi:10.1093/ehjci/jew082.

      3)     LAP also depends on the volume (intravascular volume or preload) of the patient. How the authors accounted for variations in preload and how they standardized for changes in intravascular volume among patients?

Thank you for this important comment. As you mention, LAP is affected by various causes, so we presented LAP response according to pacing as a marker. However, we could not rule out all the effects of preload change. All patients were hospitalized the prior day before the procedure, fasting for a fixed time of 8 hours. IV saline hydration at the same speed is maintained. We tried to maintain the same environment as much as possible. We revised sentence to the text.

Page 4 line 107

All patients remained fasting for 8 h prior to invasive LAP measurement. The procedure was performed under sedation using propofol.

à All patients fasted eight hours prior to invasive LAP measurement and catheter ablation. Saline fluid (0.9%) was injected during that time at a rate of 40 ml/hour without taking medication. Sedation was used during the procedure and pressure measurements.

      4)     Kaplan Meier curves (Figure 4). Number at Risk Tables should be provided. In panel A patients with low LA pressure at 150 bpm seems to have less AF recurrence. The not significant p-value might be secondary to low power (small cohort of patients). However in absence of Risk Tables it is difficult to draw any conclusion.

Thank you for this important comment. We have added number at risk table to Figure 4. In the figure, patients with higher LAPR showed an increased trend of AF recurrence. However it was not statistically significant.

      5)     Kaplan Meier curves (Figure 4). Why did the author stratify only for LA pressure at 150 bpm. What are the results for LA pressure dichotomized at other pacing cycle length? Any significant difference in AF free survival?

Thank you again. The stratification at 150 bpm was due to the largest LAP difference. Those with a high increase in LAP in 150 were also high at 120bpm and 100bpm, but there was no clear difference in all sections.

Reviewer 2 Report

Congratulations on addressing a very timely topic and an interesting study concept. I would only suggest indicating in the text the information as to the mean and SD for LAP.  If the authors have data on left atrial volume and its value indexed to BSA, this would be a more valuable parameter than just LA dimension. 

Author Response

Congratulations on addressing a very timely topic and an interesting study concept. I would only suggest indicating in the text the information as to the mean and SD for LAP.  If the authors have data on left atrial volume and its value indexed to BSA, this would be a more valuable parameter than just LA dimension. 

We deeply appreciate the many comments and the review provided by reviewer #2. These as well, significantly improved the clarity and content of the manuscript.

As you said, we added the LAP value to the text. LA dimension has been replaced with LA volume index in text and table. We reviewed LA volume data and revised all data in manuscript.

Page 5 line 160

The heart rate changed according to the right atrial pacing (RAP) interval (Figure 2). LAP did not increase at 75 bpm and 100 bpm, but rose at 120 bpm and 150 bpm. A similar pattern was observed in both the AF and control groups. LAP was not different at any of the heart rates in both groups. In the High E/e' group and the Low E/e' group, the LAPR differed according to RAP.

The heart rate changed according to the right atrial pacing (RAP) interval (Figure 2). LAP did not increase at 75 bpm, 100 bpm and 120 bpm but rose at 150 bpm (27±5mmHg in AF group; 25±4mmHg in control group). A similar pattern was observed in both the AF and control groups. LAP was not different at any of the heart rates in both groups. In the High E/e' group and the Low E/e' group, the LAP differed according to RAP (29±6mmHg vs 25±4mmHg) at 150 bpm.

Page 4 line 135, Page 5 line 149, Page 5 line 157, Table 1

LA diameter à LA volume index

Round 2

Reviewer 1 Report

Thanks for your reply. I have no further comment.